# MALDI-TOF MS: A Reliable Tool in the Real Life of the Clinical Microbiology Laboratory

**DOI:** 10.3390/microorganisms12020322

**Published:** 2024-02-03

**Authors:** Adriana Calderaro, Carlo Chezzi

**Affiliations:** Department of Medicine and Surgery, University of Parma, Viale A. Gramsci 14, 43126 Parma, Italy; carlo.chezzi@unipr.it

**Keywords:** MALDI-TOF MS, bacteria identification, fungi identification, laboratory automation, laboratory workflow, antimicrobial resistance

## Abstract

Matrix-Assisted Desorption/Ionization-Time of Flight Mass Spectrometry (MALDI-TOF MS) in the last decade has revealed itself as a valid support in the workflow in the clinical microbiology laboratory for the identification of bacteria and fungi, demonstrating high reliability and effectiveness in this application. Its use has reduced, by 24 h, the time to obtain a microbiological diagnosis compared to conventional biochemical automatic systems. MALDI-TOF MS application to the detection of pathogens directly in clinical samples was proposed but requires a deeper investigation, whereas its application to positive blood cultures for the identification of microorganisms and the detection of antimicrobial resistance are now the most useful applications. Thanks to its rapidity, accuracy, and low price in reagents and consumables, MALDI-TOF MS has also been applied to different fields of clinical microbiology, such as the detection of antibiotic susceptibility/resistance biomarkers, the identification of aminoacidic sequences and the chemical structure of protein terminal groups, and as an emerging method in microbial typing. Some of these applications are waiting for an extensive evaluation before confirming a transfer to the routine. MALDI-TOF MS has not yet been used for the routine identification of parasites; nevertheless, studies have been reported in the last few years on its use in the identification of intestinal protozoa, *Plasmodium falciparum*, or ectoparasites. Innovative applications of MALDI-TOF MS to viruses’ identification were also reported, seeking further studies before adapting this tool to the virus’s diagnostic. This mini-review is focused on the MALDI-TOF MS application in the real life of the diagnostic microbiology laboratory.

## 1. Introduction

Matrix-Assisted Laser Desorption Ionization Time-Of-Flight Mass Spectrometry (MALDI-TOF MS) is an analytical method of rapid and sensitive microbial identification and characterization based on the fast and precise assessment of the mass of ionized sample molecules by short laser pulses after their co-crystallization with a low-molecular-weight organic acid commonly referred to as matrix [1,2].

MALDI-TOF MS represents the most relevant revolution introduced in the last few years that is able to make a contribution to the clinical microbiology setting for the identification of pathogenic bacteria and fungi and the detection of resistance to antimicrobial drugs. MALDI-TOF MS is based on the proteomic analysis of the constitutive protein profiles of bacteria and fungi [3,4]. It was demonstrated to have the advantages of excellent sensitivity, high throughput, simple operation, and low cost, although the cost of the spectrometer is relatively high.

The use of MALDI-TOF MS has improved the workflow in clinical microbiology laboratories and reduced the time to obtain a microbiological diagnosis by 24 h compared to conventional biochemical automatic systems. Its application to the identification of bacteria has revolutionized the workflow in clinical microbiology laboratories and was successfully extended to the identification of fungi and, more recently, to the detection of markers related to the antimicrobial resistance mechanisms, demonstrating that it is highly throughput and very fast [3]. The application of MALDI-TOF to the identification of antimicrobial resistance markers contributing to select the appropriate preliminary antimicrobial drug before the results of the antimicrobial susceptibility test (AST) can be available. The main advantage of MALDI-TOF MS is the reduced time to obtain bacterial identification in few minutes compared to 24–48 h from the biochemical culture-based methods. The time saved in the identification of pathogenic bacteria/fungi is particularly critical in cases of life-threatening infections or in cases of slow-growing strains challenging the conventional methods [5].

Thanks to its rapidity, accuracy, and moderate price, MALDI-TOF MS has also been applied to different fields of clinical microbiology, such as the detection of antibiotic susceptibility/resistance biomarkers, the identification of aminoacidic sequences and the chemical structure of protein terminal groups, and as an emerging method in microbial typing. Most of these applications require a deeper evaluation before confirming their transfer to the routine. Due to the great advantage of its versatility, MALDI-TOF MS can be implemented by the user and adapted to the needs of the diagnostic workflow of a specific laboratory [6].

Studies were reported of the MALDI-TOF MS application to directly identify pathogens in clinical samples, producing promising results using urine samples [7] or cerebrospinal fluid [8]. The most reliable application of MALDI-TOF MS remains the smart and fast identification of microorganisms in positive blood cultures, together with the detection of antimicrobial resistance in the same run and the identification of bacteria and fungi in isolated cultures [9].

Despite the fact that several studies have been reported in the last few years on the application of MALDI-TOF MS to the identification of intestinal protozoa [10,11], helminths [12], *Plasmodium falciparum* [13] or ectoparasites [12], and *Trichomonas vaginalis* [14], MALDI-TOF MS has not yet been used for the routine identification of parasites. Innovative applications of MALDI-TOF MS to virus identification were also reported, with promising results [15,16,17].

This narrative mini-review is focused on the MALDI-TOF MS application in the real life of the diagnostic microbiology laboratory.

## 2. MALDI-TOF MS: A Short History of a Big Revolution in Clinical Microbiology

Mass spectrometry (MS) was invented in the 19th century and, at the beginning, was only used in the chemical sciences. In the 1980s, MALDI was introduced and improved the use of MS, allowing its application to identify biological macromolecules as proteins [1]. In 2002, John Fenn and Koichi Tanaka received the Nobel Prize in Chemistry for this application based on the development of soft desorption ionization techniques for the analysis of biological macromolecules, including electrospray and soft laser desorption. This technique makes the mass spectrometric analysis applicable to the large and fragile polar molecules that play vital roles in biological systems. The distinguishing features of electrospray spectra for large molecules are coherent sequences of peaks whose component ions are multiply charged, with the ions of each peak differing by one charge from those of its adjacent neighbors in the sequence. Spectra have been obtained for biopolymers including oligonucleotides and proteins, with the latter having molecular weights up to 130,000, with as yet no evidence of an upper limit [18].

MALDI-TOF MS, as a method that combines proteome analysis with C_10_H_7_NO_3_ as a matrix, was found to be beneficial for quickly identifying microorganisms and was described in 1994 by Cain et al. [2].

Since 1975, the first experiments to identify microorganisms using MS have been conducted, and although encouraging, the results were unreproducible due to the interference of growth media and culture conditions [19]. Great improvements were brought by the MALDI-TOF MS technique in the 1980s, allowing for the analysis of large biomolecules such as bacterial ribosomal proteins [20], with the advantage of poor interference by the culture medium and growth conditions and reproducible and reliable identification at the species level [21].

In the last 10 years, the implementation of MALDI-TOF MS in clinical microbiology laboratories has strongly modified the diagnostic workflow for the identification of bacteria and fungi.

In 2009, MALDI-TOF MS was widely introduced in clinical microbiology laboratories, and since then, descriptions of improvements have been reported. The most important change concerned the pretreatments of bacteria that were used at the early application of MALDI-TOF MS identification. It was demonstrated that it is possible to immediately use fresh colonies streaked on MALDI target plates with a simple preparation, and this fact quickly contributed to the practical application of the MALDI-TOF MS technology to microbial identification worldwide [20,21,22].

The milestones of the history of the MALDI-TOF MS application in the clinical microbiology are reported in Figure 1.

Bacterial identification should be ideally carried out using a fast method requiring simple and short sample preparation, automated as much as possible, producing results not requiring data analysis, and also not expensive and easily applicable to the routine [23,24]. MALDI-TOF MS is not so far from this ideal model of diagnostic device.

## 3. MALDI-TOF MS: Principles of the Procedure

Mass spectrometry (MS) directly analyzes the mass-to-charge ratios (*m*/*z*) of molecular ions, and from this, it can identify and quantify any ionized susceptible biological molecule. The mass spectrum of different molecules present in a sample can be interpreted within the spectrometer to identify each molecule [20,21,22].

MALDI requires a short pre-analytical step simply consisting of a mix prepared with an aliquot of the sample to be tested and an aliquot of a low-molecular-weight organic acid commonly referred to as matrix. After this, to generate the ionization of the molecules comprising the sample, a short laser pulse is used. The matrix protects the sample from fragmentation during the ionization. The ions generated by the actions of the laser on the molecules present in the sample are then analyzed, measuring their mass-to-charge ratios (*m*/*z*). From this, the spectra are generated and interpreted by comparing them with the spectra present in the database of the spectrometer [27]. The application of MALDI-TOF MS in clinical microbiology laboratories for the identification of bacteria and fungi uses the analysis of microbial biomolecules, mostly the highly expressed ribosomal proteins, based on the ionization and co-crystallization of the bacterial/fungal cells mixed with a matrix; then, the application of short laser pulses accelerates the ions, and their time of I confirm. flight is measured in a vacuum flight tube. The microbial molecules used for the identification are the constitutive proteins specific of each bacterial species that are analyzed using the *m*/*z* ratio to produce the protein spectra that are compared with those present in the database of the MALDI-TOF MS application. The score produced automatically from this comparative analysis allows us to assign species identification [22].

The detection limit of the MALDI-TOF MS technique is at least 10^2^–10^4^ cells, depending on the bacterial species, even if a small amount of microbial mass is enough to obtain an identification with a good score [21,22,23,26].

A tip of a bacterial colony is mixed with 1 μL of a suitable chemical matrix (e.g., a solution of cinnamic acid or a solution of a benzoic acid derivate) and dispensed on the steel target plate or on the plastic slide support to dry at room temperature. The crystallization of the sample with the matrix on the target plate is the activity required to obtain the ionization of the molecules present in the sample. When the target plate is inserted into the instrument, the vacuum needs to be reproduced and maintained during the run. As the vacuum is reproduced, the energy derived by the short laser pulses vaporizes the molecules of bacterial origin mixed with the matrix and produces their ionization. As a consequence of such a phenomenon the ionized bacterial proteins, mostly of ribosomal origin, run in the tube of flight (TOF) pushed under an acceleration derived from an electromagnetic field (about 20 kV). The time of flight is measured as the time to reach the detector situated at the end of the tube for each ionized molecule. The TOF of each protein is determined by both the mass and degree of ionization, and it is used to obtain a specific individual protein spectrum. In such a way, a biotype corresponds to a specific protein spectrum for each bacterial species, and this information is recorded to univocally identify each analyzed sample [21,22].

The time-of-flight data are used to create a distinctive protein mass spectrum that is matched to the library of the machine protein spectra referred to as the “Main Spectrum Profile” (MSP). The MSP contains signals unique to microbial genera, is species-specific to each strain, and is the fingerprint of each strain. The unknown pathogen is well-known at the genus and species levels by comparing the protein spectra of unidentified bacterial isolates to those present in the database. The range of molecular weights used for bacterial species identification is based on the constitutive ribosomal proteins and is comprised between 2 and 20,000 Da in a good signal-to-noise ratio with poor interference from microbial culture conditions. The computer software of the machine comparatively analyzes, in real time, the protein spectra of the samples tested with those present in the reference database of the machine (including specific spectra comprising over 2000 microbial species of medical interest) and produces a score (numeric value) corresponding to the similarities found. The computer software associated with the machine MALDI Biotyper (Bruker Daltonics GmbH, Bremen, Germany) uses this “score”; a score value of 2.0 is considered a reliable criterion of bacterial species identification, although the spectra obtained from a single strain are not identical in repeated runs because of a method-inherent noise. Score values between 1.7 and 2.0 are considered reliable for genus identification [22,23,24,25,27,28,29,30,31,32]. The machine MALDI VITEK MS (bio-Mérieux, Mercy L’Étoile, France) produces a confidence value, which can be reported as a percentage up to 99.9%. The return of an identification of a single taxon, regardless of the confidence value, is considered an acceptable level of identification. If no identification is provided, the isolate is considered unidentified. The Biotyper machine reports the identification of each organism with a numerical “score,” and, according to the manufacturer, scores of ≥2.0 and ≥1.7 represent acceptable probable species- and genus-level identifications, respectively [22,23,24,27,28].

Using MALDI-TOF MS, it is also possible to conduct a deeper analysis of bacterial proteins after a proteolytic digestion to obtain peptide mass fingerprinting (PMF). This approach was described for single-strain profiling and is not currently used in diagnostic practice as it requires high-resolution devices [25].

In Figure 2, the main approaches used for the applications of MALDI-TOF MS in clinical microbiology are summarized.

The great advantage of MALDI-TOF MS is the use of a small amount of bacterial growth to obtain an identification, and this can often be achieved without subcultures of the strain, using only a small part of a colony from a mixed culture or from a small colony grown after a few hours in a subculture.

The microbial colonies used for MALDI-TOF MS identification should be no more than 48 h old, as fresh cultures undergo less protein degradation and the ribosomal proteins are of better quality, ensuring a good score value for identification; on the other hand, old cultures, particularly in the case of yeasts, could have a stronger cell wall that is difficult to destroy during sample preparation, requiring a procedure of protein extraction before the mix with the matrix to obtain the ionization of the molecules.

Globally, the Microflex Biotyper (Bruker Daltonics GmbH, Bremen, Germany) and VITEK MS (bioMérieux, Marcy l’Etoile, France) are the commercially available MALDI–TOF MS systems currently implemented in clinical microbiology laboratories [28,29,30,31,32]. Recently, a new MALDI–TOF MS system, ASTA MicroIDSys (ASTA Inc., Suwon, Korea), was developed for the identification of clinically important microorganisms, producing discrepant results in the identification of microorganisms not frequently isolated in clinical laboratories, such as *Paenibacillus* spp. and *Weissella* spp. Using ASTA MicroIDSys software (version 1.27), the species identification cut-off score of 130.0 is comparable to the score of 1.7 in Biotyper, according to a report comparing the two devices [3,5,25,29,30,31]. However, the ASTA MicroIDSys does not yet have the same distribution as those of Biotyper and VITEK MS, and its evaluation by the users is not comparable to the previous ones.

MALDI Biotyper software (version number 12.0) compares each sample’s mass spectrum to the reference mass spectra present in the database using a pattern-matching approach that is based on statistical multi-variant analyses and includes peak positions and intensities; it calculates an arbitrary unit score value comprised between zero and three, reflecting the similarity between the sample and reference spectra, and finally displays the top ten matching database records. As specified by the manufacturer, identification scores ≥ 2.0 were accepted for a reliable identification at the species level and scores ≥ 1.7 and ≤2.0 for an identification at the genus level. Scores < 1.7 were considered unreliable.

The results of MALDI-TOF MS are available in less than 1 min for only one sample and in less than 1 h if a target plate with 96 samples is used. The target plates are usually reusable after a simple cleansing procedure or are disposable according to the manufacturer’s instructions (Bruker Daltonics GmbH, Bremen, Germany, or MALDI Biotyper, bioMériex, Marcy l’Etoile, France, respectively) [28,29,31].

The microbial biotype identified by MALDI-TOF MS corresponds to a fingerprint of each bacterial strain, and this is the reason supporting the introduction of such a technology in clinical microbiology laboratories, replacing—in most cases completely, in a few cases completing—the conventional method of microbial identification based on the biochemical profile analysis of bacteria/fungi subcultures grown in specific media after their isolation in pure culture. It was proven that MALDI-TOF MS is as reliable and specific as the conventional methods and more accurate for some species. The remaining advantage of the conventional methods is that they can include the AST.

## 4. Pretreatment of the Samples

Appropriate pretreatments of the sample before the analysis can make the identification of microorganisms more efficient, rapid, and accurate when used in association with the implementation of the MALDI-TOF MS database with protein spectra of different bacterial strains. There have been numerous studies reporting protocols for the pretreatment of the sample to improve the identification of microorganisms by MALDI-TOF MS, but the effect of the pretreatment on microbial identification is often neglected [23].

Protocols for sample preparation before MALDI-TOF MS analysis can influence the results of the method, and the most widely used are known to have advantages and disadvantages.

The most common simple and fast methods reducing the handling of the sample used in bacteriology are the direct colony transfer method and the extraction method directly on the target plate. The extraction method using a time-consuming treatment in a tube was more recently developed for the microorganisms that are difficult to identify, such as mucinous bacteria.

The selected protocol should fit with the routine diagnostic workflow, and the personnel involved in microbial identification should be efficient. The ideal method to prepare the sample should be fast and simple but also ensure reliable results in microbial identification for all the species involved in human diseases.

The direct colony transfer method suggested for Gram-negative bacteria consists of a tip of a bacterial colony (few amounts of biomass are enough) covered with 1 μL of an α-cyano-4-hydroxycinnamic acid (CHCA) matrix solution (a saturated solution of α-cyano-4-hydroxycinnamic acid in in 50/50 [*v*/*v*] of acetonitrile/H2O containing 2.5% trifluoroacetic acid).

Similarly, in the extraction method directly on the target plate, suggested as being better for Gram-positive bacteria, a tip of the sample is covered with 1 μL of 70% formic acid, dried at room temperature, and then overlaid with 1 μL of a CHCA matrix solution [25,33].

In both cases, after air drying at room temperature, the sample can be analyzed by MALDI-TOF MS.

For the microorganisms that have a deep cell wall (fungi, mucinous bacteria, etc.), to obtain a good ionization of their biomolecules and improve their identification, a time-consuming pretreatment has been described. Briefly, microbial colonies are treated with distilled water (300 μL) and ethanol (900 μL) in a tube and centrifuged (at 13,000× *g* for 2 min), the pellet is resuspended in 70% formic acid (15 μL), and the supernatant (1 μL) is transferred into the target plate and, after being dried at room temperature, is covered with 1 μL of a CHCA matrix solution [33]. The basic procedures of sample preparation for MALDI-TOF MS are reported in Figure 3.

The most widely used protocol allowing for the identification of the most common bacteria (Gram positive and negative) of medical interest is the extraction directly on the target plate, as it is fast and simple, only requires a small amount of sample, and thus is applicable for scarce microbial growth.

However, while it was reported that the pretreatment protocols induce changes in the protein profile of the strains, the consequences for their microbial identification are poorly investigated and often neglected.

The high sensitivity of MALDI-TOF MS allows the use of different pretreatment methods that can cause changes to the microbial protein fingerprints; consequently, the pretreatment process may have an important effect on the results, but the selection of an appropriate preparation method can improve the efficiency and accuracy of the MALDI-TOF MS identification method [33].

### 4.1. Database Implementation

MALDI-TOF MS, applied for the identification of bacteria, has revolutionized the workflow in clinical microbiology laboratories and was successfully extended to the identification of fungi and, more recently, to the detection of markers related to antimicrobial resistance mechanisms, demonstrating that it is highly throughput and very fast. The main advantage of MALDI-TOF MS is its reduced time to obtain pathogen identification in a few minutes compared to 24–48 h from the biochemical culture-based methods. The time saved in the identification of pathogenic bacteria/fungi is particularly critical for cases of life-threatening infections or in cases of slow-growing strains, challenging the conventional methods.

However, some limitations of the MALDI-TOF MS technique are not yet resolved, such as the difficulty in distinguishing between closely related species, such as *Streptococcus pneumoniae, Streptococcus mitis/oralis*, and *Escherichia coli*/*Shigella* sp. [25,30,31,34].

#### 4.1.1. Database Implementation for Gram-Positive Bacteria

The identification of Gram-positive bacteria by using MALDTI-TOF MS is less reliable compared to that of Gram-negative bacteria, and in some cases, it is still a challenge. Due to the complex chemical composition of their cell walls and their high similarity in different species and genera, the use of formic acid before protein extraction is required as an on-plate treatment before MALDI-TOF MS analysis of such bacteria. Also, Gram-positive cocci species that are strictly related to one another, such as *Streptococcus pneumoniae, Streptococcus mitis*/*oralis,* and viridans streptococci, as well as those species that are not in the database, are exceptions to the reliability and affordability of MALDI-TOF MS as a tool for microbial identification [25].

A MALDI-TOF MS assay was recently described as an application to the bile solubility test for the discrimination of pneumococci; this assay was reported as a practical implementation of the bile solubility test, overcoming the subjective visual interpretation of such a test. This assay consists of an automatic reading of the conventional bile solubility test using MALDI-TOF MS. The absence of microbial spectra (Biotyper score < 1.7) in samples identify pneumococci as result of their bile solubility that causes the removal of bacterial biomass [35].

A previous application as a scientific approach of MALDI-TOF MS for typing pneumococcal strains produced promising results, highlighting its usefulness for its rapid and cost-effective routine application in clinical laboratories. Nevertheless, extensive validation to assess the reproducibility of the results is needed and did not resolve the above-reported limitation [36].

#### 4.1.2. Database Implementation for Gram-Negative and Anaerobic Bacteria

A microorganism’s identification by MALDI-TOF MS systems is based on the database where protein spectra, mostly of the highly expressed ribosomal proteins, are recorded, and this represents an intrinsic limitation of such systems in the laboratory practice to differentiate between closely related bacterial species, i.e., *Klebsiella*, *Enterobacter*, *Citrobacter*, and *Raoultella* [25,31].

Due to the great advantage of its versatility, MALDI-TOF MS can be implemented by the user and adapted to the needs of the diagnostic workflow of a specific laboratory. This versatility stimulated studies to implement the software capacity and the database of reference spectra with those of anaerobic bacteria and fungi to obtain their reliable identification in diagnostic practice, as it was already proven for most of the Gram-negative and Gram-positive bacteria.

The use of the two commercially available and widely used MALDI-TOF MS identification systems, the Bruker MALDI Biotyper (Bruker Daltoncs GmbH, Bremen, Germany) and the Vitek MS machine (bioMérieux, Marcy L’Etoile, France), for the identification of anaerobes was extensively evaluated, demonstrating the high reliability and effectiveness of MALDI-TOF MS identification of anaerobic bacteria [33,37].

Due to the limited numbers of reference spectra stored in the reference databases, early studies on the identification of anaerobes with MALDI- TOF MS reported poor identification levels for certain species and genera. To overcome these limitations, a consortium of European expert laboratories collected and characterized clinical anaerobic isolates, particularly rare and newly described species. Subsequently, additional reference protein spectra of anaerobic bacteria were introduced into the Bruker database [33].

MALDI-TOF MS identification of anaerobic bacteria has been proven to be fast, inexpensive, and highly reliable, as it was for other bacteria. The implementation of the Bruker MALDI Biotyper system increased the number of different anaerobic genera and species, and regular updates to the MALDI-TOF MS databases are needed to keep up with the changing taxonomy of anaerobes and their growing diversity.

It has been widely demonstrated that MALDI-TOF can identify non-fermenting Gram-negative rods as well as *Pseudomonas aeruginosa*, *Stenotrophomonas maltophilia*, and *Moraxella catharralis*, and the most prevalent species of the *Enterobacteriaceae*, but misidentifies *Shigella* species as *Escherichia coli* and only partially distinguishes between *Citrobacter freundii*, *Enterobacter cloacae*, and *Salmonella enterica* species [3,25,30,31,33,34].

This limitation is also present with other species, and it is due to the similarity of some species belonging to the same genus: for example, *Burkholderia cepacia* and *Acinetobacter baumannii* can be identified only on a wide range, while *Achromobacter* species, *Chryseobacter* species, and *Ralstonia* species can only be identified at the genus level, although the differentiation of such microorganisms at the species level is not crucial in the diagnostic practice of medical microbiology.

To partially overcome the limitations related to the biological aspects of microorganisms belonging to related species, some practices are recommended: update the library; use of reference strains that are well characterized for both the genetic profile and the accurate antimicrobial susceptibility/resistance profile; implementation of a database with protein spectra from different strains to make the identification more reliable and accurate. However, when the protein spectra are derived from high protein similarity, the accurate differentiation of highly related species requires alternative methods such as genotyping and cannot be achieved by conventional methods for the identification of biochemical profiles, which is also a limitation for these methods.

The advantage derived from the implementation of the MALDI-TOF MS database is clearly represented by some examples: the successful identification of the Gram-negative bacteria of the HACEK group, including *Haemophilus*, *Aggregatibacter*, *Cardiobacterium*, *Eikenella*, and *Kingella*, the species belonging to the genus *Neisseria*, avoiding the misidentification of the commensal apathogenic species with *N. meningitidis*. However, encapsulated strains, such as *K. pneumoniae* and *H. influenziae*, can be misidentified, and the use of reference spectra implementation can help in such difficulties [6].

The benefits of database extensions by the users including the appropriate controls were demonstrated to be reliable in improving the MALDI-TOF MS technique’s diagnostic performance [32,38].

The MALDI-TOF MS technology for *Clostridioides difficile* typing was proven to be useful in supporting the epidemiological investigation performed by polymerase chain reaction (PCR) ribotyping. This was achieved by a typing MALDI-TOF MS (T-MALDI) method for the rapid classification of the circulating *C. difficile* strains in comparison with the PCR ribotyping results [39]. T-MALDI for *C. difficile* classification could be a valid alternative to PCR ribotyping. The protein spectra acquisition for MALDI-TOF MS typing in comparison with PCR ribotyping turned out to be easier (only a few manual steps and minimum hands-on time for spectra acquisition vs. several labor-intensive steps for nucleic acid amplification, DNA fragments separation, and PCR amplification pattern analysis), faster (30 min vs. at least 12 h), and cheaper (€1.5 vs. €15 for the reagents and disposable materials per strain). Moreover, T-MALDI is also suitable for a single-strain analysis, allowing for the real-time monitoring of *C. difficile*-circulating strains, whereas PCR ribotyping is optimized for the analysis of many samples in a batch.

T-MALDI also represents an alternative to PCR ribotyping in terms of its reproducibility, set up time, and costs, as well as being a useful tool in epidemiological investigations for the detection of *C. difficile* clusters involved in outbreaks [40].

#### 4.1.3. Database Implementation for Other Bacteria

Concerning pathogenic spirochetes, the MALDI-TOF MS database was implemented and applied to the identification of these bacteria.

Supplementing the existing database, limited to the sole species *B. murdochii*, with spirochaetal protein profiles, MALDI-TOF MS resulted in rapid, cheap, and reliable identification of *Brachyspira* strains at the species level, overcoming the problems previously encountered in the identification of these spirochaetes when using biochemical and genetic-based methods. MALDI-TOF MS proved to be more accurate than conventional identification methods and a reliable alternative to genetic-based methods for the identification of *Brachyspira spp*. isolates from both human and animal origins [41].

MALDI-TOF MS was demonstrated to be a powerful tool for research and diagnostics in the field of leptospirosis, with broad applications ranging from the detection and identification of pathogenic leptospires for diagnostic purposes to the typing of pathogenic and non-pathogenic leptospires for epidemiological purposes [42]. Also, for the identification of *Borrelia spp*., both for diagnostic purposes and epidemiological surveillance, MALDI-TOF MS was demonstrated to be useful [43].

The current limitations of microorganism identification by MALDI-TOF MS also include *Bordetella pertussis* and *B. bronchioseptica*, *Achromobacter xylosoxidans* and *A. ruhlandii*, *Bacteroides nordii* and *B. salyersiae*, and *Enterobacter cloacae complex* (composed of six very closely related species with similar resistance patterns: *E. asburiae*, *E. cloacae*, *E. hormaechei*, *E. kobei*, *E. ludwigii*, and *E. nimipressuralis*) [8].

In addition, *Burkholderia cereus*, *B. cepacia*, *B. mallei*/*pseudomallei*, *Achromobacter sp.*, *Citrobacter freundii*, *Enterobacter cloacae*, *Salmonella*, *Mycobacterium tuberculosis*, *M. abscessus*, and *M. avium* are among other examples of microorganisms that are difficult to identify by MALDI-TOF MS [8,25].

However, using MALDI-TOF MS, biologically related species are identified as a group, complex, or genus level, and in cases where the identification down to species/strain level is required for diagnostic reasons or clinical relevance, additional assays are required.

When the MALDI-TOF MS technique is routinely used in clinical microbiology laboratories, the implementation of a database with reference spectra is a better practice, also allowing the identification of pathogens not frequently involved in human infections. Generally, misidentification occurs not only for closely intrinsically related microbial species but also for the lack of reference spectra; this is the case reported for fungi, mycobacteria, and some fastidious bacteria belonging to the genera *Bacteroides*, *Fusarium*, and *Lactobacillus*. The implementation of the library with reference spectra, often home-made by the users, resolved the problem [5].

#### 4.1.4. Database Implementation for Mycobacteria and Fungi

MALDI-TOF MS can generally identify mycobacteria but cannot differentiate the species belonging to *M. tuberculosis complex*, and this limitation makes PCR-based assays still preferred to MALDI-TOF MS for mycobacteria identification in diagnostic practice. Bruker Daltonics GmbH (Bremen, Germany) has produced a pretreatment including crystal particles to mechanically destroy the microbial biomass, allowing ribosomal extraction that can provide identification at the species level for *Mycobacterium avium complex*, making MALDI-TOF able to achieve an accurate identification of non-tuberculous mycobacteria with a level of precision comparable to that of molecular assays. As for mycobacteria, *Actinomycetales* members such as *Nocardia* and *Streptomyces* require cell wall disruption to allow MALDI-TOF MS to correctly identify them, although due to their complex nomenclature, differentiation between similar species is still a challenge [44,45].

MALDI-TOF MS has some limitations with the identification of some groups of yeasts and fungi; however, the use of MALDI-TOF MS for the identification of *Aspergillum* sp., *Fusarium* sp., *Penicillium* sp., and dermatophytes is still poorly investigated as their identification requires cultivation where molds produce different forms such as mycelium and conidia that are hard to differentiate in the protein composition, making it difficult to obtain good protein spectra. Moreover, the in vitro cultivation of fungi is time-consuming, particularly dermatophytes, and the use of MALDI-TOF MS does not produce a relevant benefit in the diagnostic workflow in terms of time savings for their identification. This requires the development of suitable protocols to implement this tool as a reliable method for the identification of fungi [26,46].

New developments have extended the use of MALDI-TOF MS from prokaryotic organisms to eukaryotic organisms, such as yeasts, and molds, making this technique a straightforward, fast, and reliable identification method for bacteria, yeasts and molds in a cost-effective way. However, the assessment of MALDI-TOF for species-level identification of filamentous fungi is not as extensive, and studies examining the applications of this technique specifically for dermatophyte identification are limited [4,26,46].

According to previous published data [4,26], fungal identification by MALDI-TOF MS requires a chemical extraction of protein from a pure culture of the microorganisms. The pellet of mycelium needs a preliminary formic acid/acetonitrile protein extraction before adding an aliquot to the saturated α-cyano-4-hydroxy-cinnamic acid (HCCA) (2.5 mg) matrix solution (Bruker Daltonics GmbH, Germany).

MALDI-TOF MS proved to be a useful tool suitable for both the identification of fungi for diagnostic purposes and epidemiological surveillance [26]. The results of a recent study demonstrate that the MALDI-TOF MS may also be used to simultaneously classify *Candida* species and detect fluconazole-resistant strains [47].

A different MALDI-TOF MS instrument, Autof MS1000, from Autobio Diagnostics (Autobio Diagnostics, Zhengzhou, China), was recently tested for a rapid and reliable method for the identification of filamentous fungi with a new pretreatment protocol. While the Autof MS1000 instrument seems to be promising, it has a library that does not cover all filamentous fungi, and a further implementation of the database is required before validating its application to the routine [48].

A new MALDI-TOF MS ASTA MicroIDSys system (ASTA, Suwon, Republic of Korea) was developed and evaluated by comparing with both 16S rRNA sequencing and the Bruker Biotyper system [25]. The results were excellent with the identification of mycobacteria and anaerobic bacteria, such as *Peptostreptococcus anaerobius, C. difficile*, *Clostridium perfringens*, *Finegoldia magna*, and *Parvimonas micra*. The Autof MS 1000ASTA and MicroIDSys systems use a library based on an isolate-specific reference approach and are analogous in their use to the MALDI BioTyper system, while bioMérieux principles (e.g., Vitek MS) are based on taxonomical group-specific principles [25,48].

Among the MALDI-TOF devices, Biotyper was the historical pioneer for microbial identification. As a consequence of its larger field evaluation by the users, it was extensively adapted and tailored to the laboratory needs, and numerous scientific publications are available that also offer a complete evaluation of its performances in comparison with the more recently developed devices. The second commercially available device was the VITEK MS device, and for this system, the comparative studies are also towards Biotyper rather than the newly developed ones.

The commercially proposed MALDI-TOF MS devices include Shimadzu Axima (Shimadzu, Kyoto, Japan) and, more recently, the EXS2600 system from Zybio (Zybio Inc., Chongqing, China) [49,50]. The Axima device uses a software (version year 2013) tool, which, similar to the VITEK MS device, first compares all the mass fingerprints to the superspectra and, in a second step, to the individual spectra of the database. The results are expressed in percentages of similitude. Superspectra contain peaks that are common for different strains of the same species. The individual spectra correspond to the spectra of each strain cultured in specific conditions. The manufacturer recommends the validation of only the superspectra identifications with percentages comprising between 75 and 99.9%.

In comparison with the Biotyper and VITEK MS devices, these two additionally proposed devices have limited applications in laboratories worldwide and few available comparative studies. For instance, Zybio EXS2600 is not as widespread in Europe, and scarce evaluations of its diagnostic performance are available (limited to bacteria and fungi, and mycobacteria are the least analyzed) [50].

However, the few available studies reported similarities in their results of microbial identification compared to the Biotyper device, although the comparisons are limited, at present, to some fungal species and bacteria from urine samples [49,50].

### 4.2. Recent Advances in the Automation of MALDI-TOF MS

Automations applied to MALDI-TOF MS were developed for the sample preparation step to reduce the manipulation of the microorganisms. It was recently proven that automation at this level reduces the turn-around time for the microbial identification process due to the hand manipulation of microbial cultures, limiting the manual handling of the sample during the preparation of the target plate, which generally requires 30 min. This type of automation has the advantage to standardize the sample preparation procedure, including the transferring of the microorganisms from the culture plate into the target slide/plate, and subsequently add the matrix to the organisms to prepare them for the MALDI-TOF MS analysis. A recent study [51] evaluated, in three different laboratory sites, the use of the Colibrí ^TM^ instrument (Copan, Brescia, Italy) that automatically picks colonies that were previously selected by the microbiologist or by automated software and spots targets for microbial identification using either the Bruker MALDI Biotyper^®^ (Bruker Daltonics GmbH, Bremen, Germany) or the VITEK^®^ MS device (bioMérieux, Marcy l’Etoile, France). This study demonstrated that the standardized automation of the manual process of MALDI target spotting provides a benefit for the diagnostic workflow in microbial identification.

MALDI-TOF MS is a great contribution to the implementation of laboratory automation in clinical microbiology, facilitating organization and changes in the workflow with the introduction of the integrated automated device Colibrí^TM^ (Copan, Brescia, Italy). This automated step standardizes the quality of the microbial spot on the MALDI target, which is operator-independent, and improves the yield of identification with high reproducibility of the results. These results demonstrate that Colibrí is a reliable system for MALDI-TOF target preparation as well as for yeast identification, allowing increased standardization and less hands-on time [51,52,53]. This automation of the MALDI preparation step reduces human errors, avoiding manual preparation steps, and brings the advantage of a total traceability of the identification workflow.

Recently, a comparison of the performance and impact on the laboratory organization between the two most recent and widely used MALDI-TOF MS instruments—the Vitek MS Prime (bio-Mérieux, Marcy l’Etoile, France) and the MALDI-Biotyper Sirius (Bruker Daltonics GmbH, Bremen, Germany)—was published [54]. This study concluded that both systems provided a high identification rate of 97 to 98% for routine isolates despite single-spot measurements and confirmed that the quality of the identification highly depends on the purity of the cultures used, the amount of bacterial biomass smeared on the target plate, and the experience of the technicians, all of which are aspects that have been already described for the previous versions of the compared devices [54]. This study also reported that the Biotyper Sirius reference library includes 4194 species, while that of the VITEK MS IVD includes 1316 species. The VITEK MS device is based on taxonomical group-specific principles, while the MALDI Biotyper database is based on an isolate-specific reference approach [25].

Moreover, the Biotyper Sirius requires, for the identification of mycobacteria and filamentous fungi, specific kits to optimize sample preparation and additional modules to provide a comprehensive library, while the VITEK MS includes those groups. The identification rate was 97.9% for the VITEK MS Prime device and 98.9% for the Biotyper Sirius and both systems achieved 100% agreement at the genus level and 96.2% at the species level [54]. The additional modules depend on the use of a device strictly reserved for routine identification or also extended to biomarker detection, such as antimicrobial resistance biomarkers for the identification of drug-resistant microorganisms or lipid identification or investigation of colistin-resistant strains. The preparation step of the target plate was similar, and the hands-on time was 3–6 min shorter with Biotyper Sirius. The hands-on time for the measurement process was shorter for VITEK MS Prime (1.5 min/target), while the time to obtain the results was shorter for VITEK MS Prime. While both systems are very similar in terms of their diagnostic application and identification rates, their use in the laboratory routine differs slightly for target plate preparation, with the hands-on time being shorter for the Biotyper, making the application of the matrix in batches easily every 30 min instead of at each spot for VITEK MS and translating this in a smart workflow. The chemical matrix is ready to use for the VITEK MS, and it is associated with a reference *E. coli* ATCC 8739 that is used for the calibration, resulting in a rigorous calibration process that makes the hands-on time shorter for VITEK MS during the preparation, whereas the matrix is lyophilized for Biotyper. An important situation impacting on the usability of the VITEK MS is the position of the calibration spot every 16 spots that can be reused once; consequently, the operator is required to give the correct information of the origin (bacterial or fugal) of the streaked samples. If an error occurs at this level, the spot (sample and calibrator) cannot be read a second time, and a lack of identification requires additional testing with a higher level of retesting compared to Biotyper [51,52,53,54,55].

### 4.3. MALDI-TOF MS Application to Biological Samples

Recently, different studies have reported the application of MALDI-TOF MS on liquid cultures and liquid samples, such as blood, urine, or complex samples like stool.

Despite studies that have reported the potential capability of MALDI-TOF MS in identifying bacterial mixtures without requiring purification protocols, it is currently reported that the impossibility of commercially available devices in the identification of mixed cultures of microorganisms (a mixture of different bacterial species or bacteria and fungi) in biological samples directly, in liquid samples, or in stool samples [5,6].

Urine represents a suitable sample for the identification of pathogens by a direct application of MALDI-TOF MS, as it has a good intrinsic concentration of microorganisms and is usually collected in a high volume (up to 15 mL), facilitating sample preparation and obtaining a good biomass by a centrifugation step. The few studies published reported the identification of pathogens from urine samples with good concordance compared with traditional methods, although with a large range of agreement ranging from 79 to 92% limited to monomicrobial infection. A pretreatment protocol of urine samples by centrifugation and filtration produced disappointing results, as only 58% of identification was achieved due to the interference of antimicrobial peptides from human origin with MALDI-TOF MS and reaching 67% of the correct identification rate when 1 h diafiltration was used. However, such methods, although they produce the identification of uropathogens in less than 1 h, are laborious, not standardized, and require well-trained personnel as they are not automated and not easily applicable for diagnostic laboratory routines [7].

On the contrary, better results on the identification of uropathogens were obtained with the short-growth method of incubating for 5–6 h streaked urine samples on agar plates and harvesting the grown film to obtain a biomass sufficient for MALDI-TOF MS identification. Considering that plating urine samples is mandatory for colony counting and the AST, this method could be implemented in the laboratory workflow. In addition, this short-growth method could be automated using a plate inoculation robot (WASP^®^ by Copan, Brescia, Italy; PREVI^®^ Isola by bio-Mérieux, Marcy l’Etoile, France, or InoqulA^®^ by BD Kiestra, Franklin Lakes, NJ, USA) making it feasible in high-routine laboratories [7].

Few studies have investigated the application of MALDI-TOF MS for the fast identification of microorganisms from cerebrospinal fluids (CSFs) in cases of meningitis with a short preparation protocol: the CSF (500 μL) was mixed with 13% SDS (100 μL) and centrifuged (at 13,000× *g* 2 min) to pellet the microbial biomass that was then washed in water (1 mL) by centrifugation in the same condition and resuspended in ethanol formic acid for protein extraction and then processed with MALDI-TOF MS. These results are encouraging and suggest a deeper evaluation of MALDI-TOF MS in routine diagnostics to be confirmed [8,55].

While some major problems remain, MALDI-TOF MS can be directly used for the identification of bacteria from positive blood cultures (BCs) [9]. When positive BCs are due to the presence of polymicrobial communities, false identification can occur, and the limitations reported for isolated bacterial species also remain when they are present in BCs, such as viridans streptococci and encapsulated strains of *Klebsiella pneumoniae* and *Haemophilus influenziae*. Additional difficulties in achieving microbial identification in BCs are caused by the presence of human blood cells that interfere with generating noise spectra if not previously removed [3].

The benefits of an early pathogen identification directly from positive BCs include the earlier de-escalation and/or administration of a targeted antimicrobial treatment, improving the patient outcome and supporting antimicrobial stewardship [7,28].

MALDI-TOF MS technology has been widely investigated for the direct application to positive BCs to speed the turn around time. Starting with the positive BCs, many procedures and protocols were developed over the years. All these strategies can be grouped into two approaches: short subcultures based on identification after a short incubation period (2–6 h) of a plate subculture of the BCs, and identification directly from the BCs [7]. The first approach showed good results with fast-growing microbial species; on the contrary, they are intrinsically not suitable for slow-growing and fastidious bacteria [7]. The second one, despite requiring a sample pre-treatment to obtain a purified bacterial pellet, is significantly faster than short subcultures, and it is enabled to virtually identify every microbial species present in the MALDI-TOF MS database [7,56]. An improved diagnostic workflow for the BCs using MALDI-TOF MS is reported in Figure 4.

To achieve a purified bacterial pellet suitable for direct identification by MALDI-TOF MS, many different reagents and protocols, home-made or commercial, have been described [7,57]. The use of saponin or ammonium chloride have been described to facilitate the lysis of blood cells [57]. The centrifugation–filtration approach, employed to remove cells and cellular debris, was found to be more efficient in identifying Gram-negative bacteria, while limited data were obtained for Gram-positive microorganisms. Additional in-house methods requiring several centrifugation steps for the separation of blood cells from microorganisms are proposed; they can be performed with differential centrifugation but also with gel separator tubes by a simple and cheap method. However, to obtain a consistent microbial pellet, some methods started with a larger blood culture volume, but the extraction procedure does not always lead to a better accuracy.

MALDI-TOF MS is also currently used as a technique independent from culture conditions due to its simple and concise protocols of treatment, which allow cell lysis and immediate treatment with the chemical matrix.

Different pretreatment protocols were developed to apply MALDI-TOF MS in the laboratory diagnostic practice to the identification of positive blood cultures bottles, allowing for the lysis of the blood cells to obtain a cleaned bacterial/fungal pellet to be used for the identification. These protocols are standardized in ready-to-use commercially available kits. The MALDI-TOF Sepsityper^TM^ workflow (MSW), a commercial in vitro diagnostic kit (Sepsityper) in association with the Biotyper System (MBT) for the identification of positive BCs, can identify pathogens from positive blood cultures within 15–20 min in a cost-effective, efficient workflow that allows us to improve patient care [7].

Several studies have demonstrated that this kit allows excellent identification for Gram-negative and Gram-positive identification at the genus level, but it is less performant for identification at the species level, remaining at 91.4% for Gram-negative and 67.7% for Gram-positive species identification. Moreover, problems in the identification of anaerobic bacteria and polymicrobial pathogens from positive BCs were reported [7,38].

The most used method for the identification of bacteria and fungi from blood cultures is the Sepsityper^®^ kit. It is a commercially available kit suitable for use with the Bruker MALDI Biotyper instrument using a simple protocol that allows us to obtain a microbial pellet as biomass for MALDI-TOF MS identification directly from positive blood cultures with centrifugation and washing out of blood cells [7,8,55].

Bruker offers an in vitro diagnostic use (IVD)-marked solution with the MBT Sepsityper IVD Kit, while the VITEK MS Blood Culture Kit is for research use only [52].

The pretreatment to remove proteins of non-bacterial origin is required as a prerequisite for the successful application of MALDI-TOF MS to the identification of microorganisms from BCs. This newly proposed protocol uses 1 mL of positive blood culture broth directly in a 30-min sample preparation and MALDI-TOF MS identification, saving at least 24–48 h compared with the time of conventional identification of positive blood culture samples by MALDI-TOF MS using bacterial colonies grown on agar plates.

The evaluation of the systems ASTA MicroIDSys, Shimadzu Axima (Shimadzu, Japan), and EXS2600 from Zybio (Zybio Inc., Chongqing, China) applied to biological samples are not yet reported, while the kits for BCs have been reported in a comparison of the performances between AutoMs1000 (Auobio Diagnostics, Zhengzhou, China) and EXS2600 (Zybio Inc., Chongqing, China) [50].

### 4.4. Limitations of MALDI-TOF MS

Some limitations in the use of MALDI-TOF MS in the clinical microbiology laboratory are present and require further targeted developments and deeper technological improvements. They can be summarized as follows: The first limitation of MALDI-TOF MS is that, despite its high sensitivity to obtaining identification at the species level, the biomass should be composed of between 10^2^ and 10^4^ cells, according to the microbial species. This often requires a sub-cultivation for the slow-growing microbial species.

The second most important limitation is the pretreatment required for the identification of fungi/molds and mycobacteria. This additional step in the preparation of the samples, even if it could be automated by suitable modules completing the main device, requires additional time and skilled personnel in the laboratory to achieve microbial identification. This also increases the cost of the apparatus required in the diagnostic workflow and does not allow the elimination of conventional assays to identify these microorganisms. Moreover, the use of fresh cultures is mandatory for the identification of fungi/molds and mycobacteria. On the other hand, the use of pretreatment protocols to improve these limitations have not been investigated enough.

Among the most important limitations of MALDI-TOF MS is the impossibility of the commercially available devices to identify and differentiate mixed cultures of microorganisms (a mixture of different bacterial species or bacteria and fungi) either in cultures in vitro or in biological samples. Moreover, currently, it is not yet possible to directly apply MALDI-TOF to biological samples. In addition, isolated microorganisms in pure cultures belonging to biologically related species cannot be discriminated against, and additional tests are required. This is the case among the Gram-positive bacteria: for pneumococci and oral streptococci, there are several species among the enterococci. Similarly, among the Gram-negative bacteria, differentiation cannot be achieved between *Escherichia coli*/*Shigella* and among *Salmonella* enterica strains for *Citrobacter freundii* and *Enterobacter cloacae*. The same problem is also present for the identification of the species belonging to *Mycobacterium tuberculosis complex*, *M. abscessus*, and *M. avium.*

It cannot be neglected that the cost of the device is still a limitation for the introduction of MALDI-TOF MS in the routine of a limited resource setting, considering that the use of conventional assays cannot be totally replaced.

### 4.5. Detection of Antimicrobial Resistance

The detection of the resistance mechanisms of the antimicrobial drugs using MALDI-TOF MS represents a complementary tool to the standard AST, allowing for the prompt identification of the isolates with a mechanism of resistance at least 24 h sooner than the AST. The possibility of identifying the bacteria/fungi in association with the detection of bacterial antimicrobial resistance markers using MALDI-TOF MS makes it a fast and reliable tool in the routine of the clinical microbiology laboratory. The application of MALDI-TOF MS for the identification of all the possible antimicrobial resistance mechanisms was described using different approaches mainly based on: (i) the analysis of antimicrobial molecules and their bacterial modified products; (ii) the analysis of specific components of the bacterial cells; (iii) the analysis of bacterial ribosomal DNA methylation; and (iiii) the detection of mutations with mini-sequencing [58].

In these analyses, MALDI-TOF MS was proven to provide results useful for diagnostic purposes, whereas some of these applications represent methods that are only useful in reference centers or applicable in research laboratories [58].

The introduction of MALDI-TOF MS as a reliable tool for the fast detection of bacterial antimicrobial resistance markers allowed us to improve a prompt governance of empiric therapy in critical infections.

The methods of using MALDI-TOF MS in such a manner allow the detection and identification of bacterial strain-specific markers or the detection of products derived from bacterial drug degradations. An additional way still requiring a deeper evaluation before transferring its application to diagnostic practice is the detection of non-radioactive (stable) isotope-labeled amino acids.

The availability of smart and fast solutions able to detect antimicrobial resistance, particularly in health care systems, might contribute to limiting the prevalence of multi-drug-resistant bacterial strains.

Commercial solutions are available for detecting antimicrobial resistance from bacteria in positive blood cultures using reagents and modules combined with the MALDI Biotyper^®^ MALDI-TOF MS device.

The MBT Sepsityper^®^ workflow was demonstrated to be able to detect bacterial resistance against first-line antimicrobials. The bacterial cells isolated from positive blood cultures can be used in the MBT STAR^®^-BL IVD assays for beta-lactamase activity detection. Similarly, the MBT STAR^®^-Carba IVD Kit, in conjunction with the MBT STAR^®^-BL IVD software module, allows for the rapid identification of bacteria and detection of carbapenemase activity in one workflow. In addition, by means of the easy-to-use MBT STAR^®^-Cepha IVD Kit, cephalosporinase activity towards third-generation cephalosporins can be detected within one hour from a positive blood culture signal.

The MBT STAR^®^-Carba IVD assay is specific for the rapid detection of prevalent Class *Acinetobacter* spp. The assay can be used for microorganisms derived from culture plates and from positive blood cultures using the MBT Sepsityper^®^ IVD Kit to rapidly identify bacteria and their carbapenemase activity on the MALDI Biotyper^®^ IVD in the same workflow [25,59,60].

Bacterial antimicrobial resistance mechanisms include the hydrolysis degradation mechanism of antimicrobial drugs. The derived products show a different molecular mass from that of a native molecule. Using a specific sample preparation protocol, MALDI-TOF MS allows for the analysis of antimicrobial drugs and their degradation products that are smaller than 1000 Da [25,59,60].

Several studies have shown that MALDI-TOF MS can be used to identify carbapenem resistance and colistin resistance among *Enterobacterales*.

Carbapenem-resistant *Enterobacteriaceae* (CRE) represent a serious and growing threat to public health. The introduction of rapid and sensitive methods for the detection of carbapenemase-producing bacteria is of increasing importance. Carbapenemase production can be detected using non-molecular methods (such as the modified Hodge test, the synergy test, the Carba NP test, and the antibiotic hydrolysis assays) and DNA-based methods. A modified version of a previously described meropenem hydrolysis assay (MHA) by MALDI-TOF MS for phenotypic detection in 2 h of carbapenemase-producing *Enterobacteriaceae* was described [61]. The MHA was successfully applied to detect carbapenemase activity in well-characterized *Enterobacteriaceae* strains, producing KPC or VIM carbapenemases, and in carbapenem fully susceptible strains. This assay, also applied to NDM- and OXA-48-producing strains and to CRE with resistance mechanisms other than carbapenemase production, has been proven to be able to distinguish between carbapenemase-producing and non-producing *Enterobacteriaceae*. The MHA by MALDI-TOF MS analysis is independent from the type of carbapenemases involved, and it is faster and easier to perform/interpret than culture-based methods [60]. On the other hand, it cannot detect other carbapenem resistance mechanisms, such as porin alterations and efflux mechanisms [61,62]. This form of antimicrobial resistance is mostly related to the misuse and overuse of antimicrobials, which led to the emergence of multidrug-resistant (MDR), extensively-drug-resistant (XDR) and pan-drug-resistant bacteria. Infections caused by Gram-negative resistant bacteria, such as *Enterobacteriaceae*, *Pseudomonas aeruginosa*, and *Acinetobacter baumannii,* are a broad matter of concern because of the ineffectiveness of conventional treatments and the lack of new antimicrobial agents against them. Therefore, the occurrence and spread of resistant bacterial strains prompted the re-evaluation of polymyxins (colistin and polymyxin B), an old class of cationic, cyclic–polypeptide antibiotics, whose clinical use was previously limited for their reported nephrotoxicity and neurotoxicity. Colistin is considered a “last resort” antibiotic, namely a valid alternative to the classic antimicrobial agents ineffective against MDR Gram-negative pathogens. Given its saving role against the life-threatening MDR and XDR bacterial infections, colistin was largely and recklessly employed in both human and veterinary medicine, resulting in the emergence of colistin-resistant pathogens, mainly Gram-negative bacteria [62].

The MALDI-TOF MS approach for testing polymyxin resistance is based on the detection of biomarkers associated with the modified lipid A, which is the phenotypic result of both chromosomal and plasmid-encoded resistance to colistin in Gram-negative bacteria. Therefore, given the inherent negative charge of lipid A, several studies aimed to create MALDI-TOF MS tests to screen colistin resistance in Gram-negative bacteria by operating in a negative ion mode of the mass spectrometer. However, to date, the negative ion mode is not currently and widely available on diagnostic routine mass spectrometers, since it works in a molecular mass range different from that used for bacterial and fungal identification. An alternative approach for the identification of colistin resistance in Gram-negative bacteria by a MALDI-TOF MS protein peak-based assay was described. It was developed based on spectra acquired in a positive linear mode embedded in the most widely used MALDI-TOF MS instrument available in clinical microbiology laboratories. A classifying algorithm model (CAM), which is simple, fast, and inexpensive, was developed to rapidly detect and identify colistin-resistant strains in clinical practice [60,62].

Applications of users’ developed protocols to MALDI-TOF MS for antimicrobial detection are still ongoing.

MALDI-TOF MS is now widely used in clinical microbiology for bacterial identification, taxonomy, and strain typing. Several approaches have been proposed in MALDI-TOF MS to detect antimicrobial resistance, such as the detection of the entire cell profile, enzymatic activity by antibiotic hydrolysis, or resistance proteins within the cell. Several potential biomarkers of drug-resistant genotypes in *S. aureus, A. baumannii, P. aeruginosa*, and *K. pneumoniae*, as well as hypervirulence in *C. difficile*, using a direct approach were described [60,61,62,63,64,65].

Recently, a machine learning (ML) algorithm recently applied to MALDI-TOF was demonstrated to be useful to detect colistin resistance in *K. pneumoniae* and methicillin-resistant *Staphylococcus aureus* (MRSA) [60,64].

Similarly, the application of the MBT Lipid Xtract^TM^ Kit in combination with MALDI Biotyper^®^ Sirius dedicated software (library version 12.0) was reported to be able to accurately identify colistin resistance in *E. coli* isolates [59,65].

In Table 1, the main properties and applications of the commercially available MALDI-TOF MS devices are summarized.

## 5. Innovative Applications for Parasite and Virus Identification

MALDI-TOF MS has already revolutionized the identification of bacteria and fungi; nevertheless, it has not yet been used for the routine identification of parasites. Studies have been performed in the last few years reporting the use of MALDI-TOF MS for the identification of intestinal protozoa either by detecting specific biomarkers, such as in the case of *Cryptosporidium* spp. [66], *Giardia* spp. [67], *Entamoeba histolytica*/*Entamoeba dispar* [11], and *Dientamoeba fragilis* [10], or by creating a specific protein profile, such as in the case of *Blastocystis hominis* [68] and *Trichomonas vaginalis* [14].

MALDI-TOF MS was reported as a method to potentially detect and quantify *Plasmodium falciparum* in human blood in a recent published study [13]. It provided a proof-of-concept for the MALDI-TOF-based diagnosis of human malaria; however, while it represents an explorative attempt to use MALDI-TOF MS as a diagnostic tool for malaria, it is far from a practical application.

MALDI-TOF MS applications for parasite identification require deeper investigations before being extended to routine application; parasite identification using MALDI-TOF MS has limited application as the use of complex liquid media, such as that used for the cultivation of intestinal protozoa, interferes with the creation of a species-specific protein profile in contrast to what is normally conducted for bacteria and fungi, which grow on solid and axenic media.

The lack of reference protein spectra in the databases of the devices is the main obstacle to parasite identification, including helminths [12]. In addition, MALDI-TOF MS analysis of stool samples requires preliminary protocols for eggs/larvae purification to create stage-specific spectra for each parasite that can be used as a reference. MALDI-TOF MS is promising to be used for parasite identification, but its application in this field is still a challenge; however, it has also been applied for identification of ectoparasites such as ticks, lice, fleas, or bed bugs [12].

Pilot studies suggested the application of MALDI-TOF MS to virus identification: polioviruses [15], poliomavirus JC virus [16], respiratory viruses [16]; the results suggest MALDI-TOF MS as a potential alternative or complement to conventional diagnostic methods in virology.

The ability of MALDI-TOF MS to successfully identify strain variants has been achieved for JCV using an innovative approach applied to viruses [16]. MALDI-TOF MS was used for the measurement of nucleic acid sequence variations. Briefly, sequences amplified by polymerase chain reaction are subjected to in vitro transcription and base-specific RNA cleavage. The mass of the cleavage products produces a unique fingerprint of the sample that is compared to a calculated list of molecular weights derived from an in-silico digest of reference sequences generated via the NCBI website. Sequenom’s MassARRAY iSEQTM software is used to identify the best-matching reference sequences based on a confidence score and sequence variation probability [16]. This approach was derived from the application of iSEQTM to the identification, subtyping, and mutation detection of *Mycobacterium* spp. and *Neisseria meningitidis* [16]. The Sequenom’s MassARRAY method was also applied for large-scale detection of all known human herpesviruses in a wide variety of archival biological specimens [69].

Applications of MALDI-TOF MS for virus identification, in particular those with diagnostic purposes, couple this technique with PCR, often resulting in very expensive procedures. The development of a MALDI-TOF MS genotyping assay suitable for detecting Hepatitis B virus (HBV) variants in a sensitive and specific manner was published. The assay is based on PCR amplification and mass measurement of oligonucleotides containing sites of mutation. The MALDI-TOF MS-based genotyping assay was sufficiently sensitive to detect as few as 100 copies of the HBV genome per milliliter of serum, with superior specificity for determining mixtures of wild-type and variant viruses. When sera from 40 patients were analyzed, the MALDI-TOF MS-based assay correctly identified known viral variants and additional viral quasi-species not detected by previous methods, as well as their relative abundance. The sensitivity, accuracy, and amenability to high-throughput analysis make the MALDI-TOF MS-based assay suitable for mass screening of HBV-infected patients receiving lamivudine and can help provide further understanding of disease progression and response to therapy [70].

A recent attempt to apply MALDI-TOF MS to the identification of SARS-CoV-2 as a model of enveloped viruses was described [71]. However, this application to current diagnostic protocols in viral identification is quite far from reliable, as pretreatments allowing enrichment of viral particles and innovation in bioinformatics and MALDI-TOF MS devices need to be developed and tested.

The application of MALDI-TOF MS to virus identification and the detection of specific viral biomarkers focused on differentiating between virus-infected and uninfected cells, thus extending the use of this technology to a novel application. However, generally, the analytical sensitivity of MALDI-TOF MS is not enough to detect and identify viruses before their amplification in culture [15,16]. This is the main difficulty in applying MALDI-TOF MS directly to biological samples.

One of the main advantages of this approach is that the identification of poliovirus strains by MALDI-TOF MS analysis can be obtained after a 5-day procedure (starting from the cytopathic effect observation upon sample cultivation), significantly shortening the time needed to perform the gold standard neutralization test (about 20 days). This confirms the ability of the MALDI-TOF MS platform to significantly shorten the times for conventional identification methods in some specific cases (in the case of polioviruses, the neutralization assay) and, moreover, the significant reduction of reagent costs and the need for experienced personnel [15].

However, despite several approaches reported, such as the detection of PCR amplicons, peptide fingerprints/proteotyping, host–response profiling, and viral protein detection, in the field of clinical virology, MALDI-TOF remains only a promising technology for routine diagnosis, requiring further developments.

### Comparison with Conventional Biochemical and PCR-Based Assays and Future Perspectives of MALDI-TOF MS

In comparison with other assays currently used in clinical microbiology laboratories, such as real-time PCR (RT-PCR) and biochemical assays, MALDI-TOF MS has been a revolution and a game-changer when used as the primary identification assay. The most remarkable differences between MALDI-TOF MS and conventional assays used for microbial identification are time and cost per sample. The technician’s working time is drastically reduced in preanalytical procedures as well as the turnaround time due to automated analytical procedures to obtain the result. This produces substantial rapidity in obtaining a diagnosis.

The cost of the MALDI-TOF MS device is comparable to that of other equipment used in diagnostic laboratories, such as RT-PCR devices, but the cost per sample is significantly cheaper. Compared to conventional biochemical identification, the time with the MALDI-TOF MS is reduced to minutes from days, and the hand manipulation by the laboratory personnel is reduced to minutes from hours. This is a great advantage in the identification of anaerobes and fastidious microorganisms.

When using both conventional and PCR-based assays, it is challenging to distinguish between organisms with similar phenotypes, biochemical characteristics, and genetic characteristics. The use of protein fingerprinting introduced with the use of MALDI-TOF MS has also improved the accuracy of microbial identification based on the PCR method. Although excellent results can be obtained with both approaches, MALDI-TOF MS in a single run can provide identification down to genus, species, and strain level independently from the occurrence of mutations in the target detected and can discriminate better than biochemical assays, as in the case of HACEK isolates. The accuracy of the identification achieved by the MALDI-TOF MS device depends on the quality of the database. On the contrary, PCR-based assays fail in identification when mutations occur in the target used or when high homology in the target sequence hinders discrimination among closely related strains/species requiring additional steps such as sequencing or restriction length polymorphism analysis to achieve identification [22,25]. On the other hand, MALDI-TOF MS does not allow the AST that can be obtained in combined assays with biochemical identification.

Compared to the PCR-based assays, the use of MALDI-TOF MS requires less intensive training of the laboratory personnel and does not require dedicated areas preventing nucleic acid/amplicon contamination. These PCR-based methods are expensive and time-consuming (requiring hours for technicians’ manipulation in the preanalytical steps and turnaround time for automated analytical procedures to obtain the result) and suffer from technical limitations such as contamination and the presence of inhibitory compounds. PCR-based assays require specific reagents for a specific target in a single run and are specific for a single species/strain. For these reasons, such assays in clinical microbiology laboratories are nowadays reserved for the identification of a minority of isolates and are still preferred for the identification and genetic drug resistance detection of mycobacteria. Similar considerations can be made in comparing MALDI-TOF MS with the 16SrRNA next-generation sequencing (NGS) technique mainly used for analyzing the microbiome. NGS is not yet widely used in the microbial identification routine, not only for the cost, which is very expensive, but also for the scarce evaluation in clinical microbiology practice and the lack of standardization. Moreover, NGS competence in bioinformatic analysis is required, together with that in clinical microbiology, and this is yet far from the routine application in diagnostic microbiology laboratories.

The future perspectives of MALDI-TOF MS could be summarized as follows: (i) The development of future proteomic-based techniques and bioinformatic facilities, together with user-implemented or updated databases and libraries, could enhance MALDI-TOF MS and resolve the disadvantages related to the low discrimination of closely related species. (ii) The implementation of MALDI-TOF MS in bacterial lipid analysis could bring the possibility of extending MALDI-TOF MS application to antimicrobial resistance detection and to deeper discrimination at the strain/serotype level. The possibility of identifying microbial lipids could allow the application directly to biological samples and body fluids, avoiding the step of microbial growth in culture media. (iii) Progress in sample preparation using optimized procedures could make the identification of mycobacteria and fungi easier.

A stimulation future perspective is the development of a MALDI-TOF MS–molecular-based technique without the involvement of sequencing to retrieve genetic information about the microbial strain identified. This innovative approach could allow the identification of microbial clusters responsible for hospital infections or the detection of virulence markers or bacterial typing without the need for molecular-based assays. Due to its high throughput, speed, sensitivity, and accuracy, the future of MALDI-TOF MS implementation in clinical microbiology laboratories can be described as being used in conjunction with other leading-edge techniques, such as molecular ones, bringing significant impacts to the diagnosis and therapy of human infection [5]. MALDI-TOF MS imaging is appearing as a powerful tool for the visualization of microbial metabolites in biofilms [6]. This innovative MS-based metabolomic technique permits the direct visualization of the spatial distribution of microbial metabolic signals in the sample by collecting the spectra in specific locations [25]. This application could allow the study of the pathogen–host interaction in vivo, including the study of small molecules such as antimicrobial drugs, and correct taxonomic classification [25].

## 6. Conclusions

MALDI-TOF MS is a smart and reliable technique that has been implemented in clinical microbiology laboratories to replace or complement conventional phenotypic identification of bacteria and fungi. MALDI-TOF/MS allows us to reduce the turn-around times by an average of 1.45 days in comparison with the traditional phenotypic methods used for microbial identification. The automation of the manual process of MALDI target spotting brings additional benefits to the diagnostic workflow in microbial identification as it reduces human errors by avoiding manual preparation steps and permits a total traceability of the identification workflow. Commercially available ready-to-use reagents and their use in combination with specific modules to implement the capability of the instrument or extend its database for microbial identification are reliable in the diagnostic routine. Their selection and use should depend on the volumes of activity in the specific laboratory and on the application of the MALDI-TOF MS device, strictly reserved for routine identification or extended to the user’s research necessity. The detection of the resistance mechanisms of the antimicrobial drugs using MALDI-TOF MS represents a complementary tool to the standard AST, allowing the prompt identification of the isolates with a mechanism of resistance at least 24 h sooner than the AST. The possibility of identifying bacteria/fungi in association with the detection of bacterial antimicrobial resistance markers using MADLI-TOF MS makes it a fast and reliable tool in the routine of clinical microbiology laboratories.

## Figures and Tables

**Figure 1 microorganisms-12-00322-f001:**
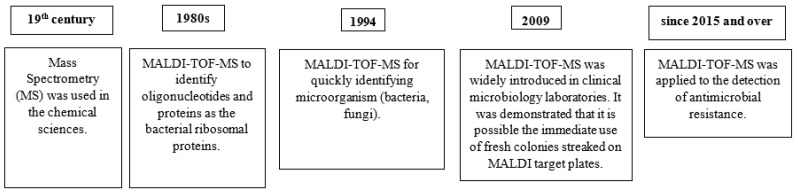
Milestones of the history of the MALDI-TOF MS application in clinical microbiology [1,2,18,20,21,22,23,24,25,26].

**Figure 2 microorganisms-12-00322-f002:**
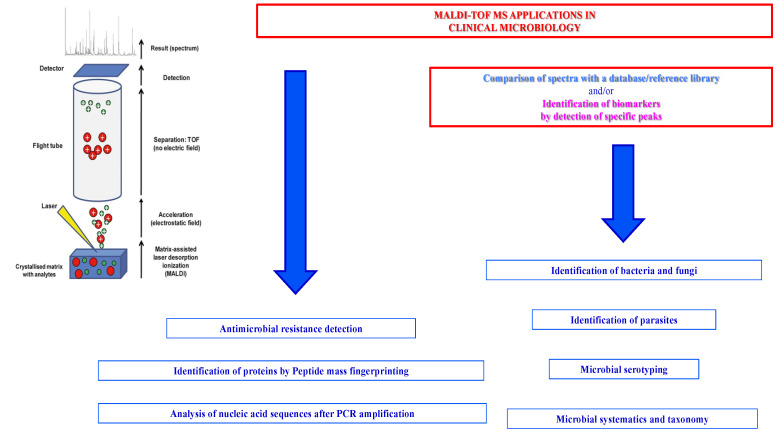
MALDI-TOF MS: the main approaches used for the applications in clinical microbiology. The approaches used for the application of MALDI-TOF MS in clinical microbiology include the comparison of the protein spectra of the sample with a database/reference library and/or the identification of specific biomarkers in the sample. Both approaches, also in combination, allow the identification of microbial proteins by peptide mass fingerprinting and the analysis of nucleic acid sequences and amplification products. The resulting applications in clinical microbiology include microbial identification, taxonomy, antimicrobial resistance detection, and microbial serotyping. Both approaches were also described for the identification of parasites.

**Figure 3 microorganisms-12-00322-f003:**
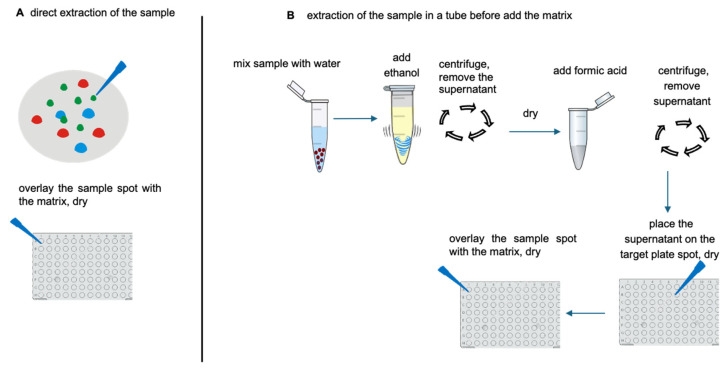
Basic steps of sample preparation.

**Figure 4 microorganisms-12-00322-f004:**
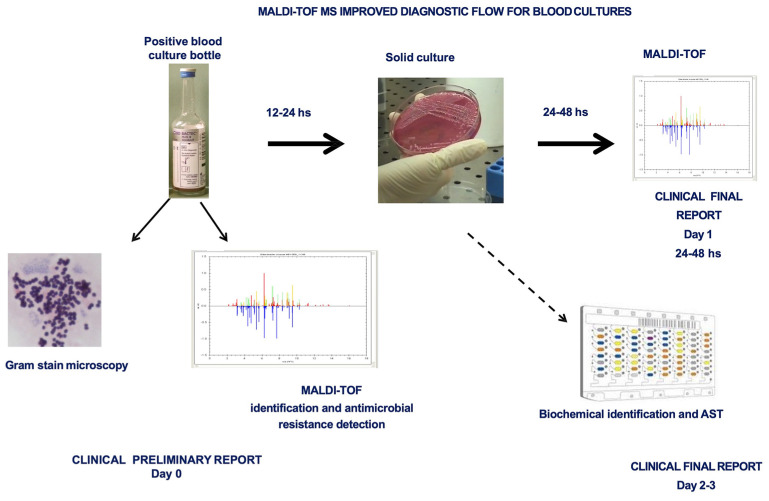
MALDI-TOF MS improved the diagnostic workflow for blood cultures: the first clinical report is available a few hours (day 0) after a positive blood culture signal; the clinical report of microbial identification after a subculture in solid medium is available after 24–48 h from the positive blood culture signal; the final clinical report after biochemical identification with a conventional assay and AST of an isolated strain is available after 2–3 days from the positive blood culture signal.

**Table 1 microorganisms-12-00322-t001:** Main properties and applications of the commercially available MALDI-TOF MS devices in clinical microbiology.

Property/Application	Biotyper Sirius (Bruker Daltonics, Bremen, GmbH, Germany)	VITEK MS (bioMérieux, Mercy Étoile, France)	ASTA MicroIDSys (ASTA Inc., Suwon, Korea)	Autof MS1000 (Autobio Diagnostics, Zhenhzhou, China)	Shimadzu Axima (Shimadzu, Japan)	EXS2600 (Zybio Inc., Chongqing, China)	Reference
Sample preparation							[23,33,50,51,52,53]
Manual	Yes	Yes	Yes	Yes	Yes	Yes	
Automatable	Yes (Colibrì^TM^ instrument)	Yes (Colibrì^TM^ instrument)	No data	No data	No data	No data	
Sample pretreatment	No	For mycobacteria and fungi	For mycobacteria and fungi	For mycobacteria and fungi (FA)	For mycobacteria and fungi (FA)	For fungi (FA)	[29,50]
One step ready-to-use matrix (type)	No (CHCA matrix lyophilized)	Yes (CHCA matrix for bacteria; FA matrix for fungi)	No (two steps: FA, followed by the CHCA matrix for all use)	No (two steps: pretreatment kit, followed by the CHCA matrix)	No data	Yes	[33,37,50]
Target plate							[25,48,50,51,52,53,54]
Capacity	96 spots	48 spots	96 spots	96 spots	96 spots	96 spots	
Disposable	No	Yes	Yes	Yes	No	No	
Reusable	yes	No	No	No	Yes	Yes	
Database for microbial identification and microorganisms that have been identified	4194 species (BMT IVD reference library 12.0) based on isolate-specific references approach; Gram-positive Gram-negative Anaerobes Mycobacteria not included Filamentous fungi not included	1316 species (VITEK IVD 3.2 knowledge base) based on taxonomical group-specific principles; Gram-positive Gram-negative Anaerobes Mycobacteria included Filamentous fungi included	3193 species (ASTA library Core DB version 1.27-biuld001) based on isolate-specific references approach; Gram-positive Gram- negative Anaerobes Mycobacteria included Filamentous fungi included	Over 5000 species (libray v1.1.0) Gram-positive Gram- negative Anaerobes Mycobacteria included Filamentous fungi included	About 5000 species (database year 2013) Gram-positive Gram- negative Anaerobes Filamentous fungi included; for mycobacteria: no data	4051 species (V.1.0.0.0 database) Gram-positive Gram- negative Anaerobes (Mycobacteria included: few data available) Filamentous fungi included	[29,32,33,37,38,40,48,49,50,53]
Identification criteria	Pattern matching approach based on “Main Spectrum Profile”; similarity expressed as “log(scoring)” (number) to peak patterns in the data base	Algorithm based on machine learning “Advanced Spectra Classifier”; Confidence values (%) for the similarity to a reference species in the database	Analogous to MALDI Biotyper	Analogous to MALDI Biotyper	Analogous to VITEK MS	Analogous to MALDI Biotyper	[25,49,50,53]
Users’ database implementation	Yes	No	No	No	No data	No data	[5,10,11,14,26,32,41,42,43]
Additional modules	MTB Mycobacteria IVD module MTB HT filamentous fungi IVD module MTB Subtyping IVD module MBT HT Spsityper IVD module MBT STAR-BL IVD module (for beta lactamase detection)	No	No	No data	No data	No data	[7,25,57,58,59,60,61,62,63,64]
Additional kits	MTB Sepsityper IVD kit (can be associated with MBT STAR-BL IVD module)	VITEK MS Blood Culture Kit (Research Use Only) VITEK MS Mould Kit (IVD) VITEK MS Mycobacteria/Nocardia Kit (IVD)	No data	AUTO MS pretreatment kit for blood cultures	No data	Manual kit for the pretreatment of blood cultures	[7,25,50,57,58,59,60,61,62,63,64]
Antimicrobial detection	MTB STAR-Caba IVD assay (associated with MBT STAR-BL IVD module) MTB STAR-Cepha IVD assay	No data	No data	No data	No data	No data	[7,25,57,58,59,60,61,62,63,64]

## Data Availability

The data presented in this study are available in the manuscript.

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
