# Peer review of "MALDI-TOF MS: A Reliable Tool in the Real Life of the Clinical Microbiology Laboratory"

_microorganisms, 2024, doi:10.3390/microorganisms12020322_

Round 1

Reviewer 1 Report (Previous Reviewer 1)

Comments and Suggestions for Authors

The manuscript titled "MALDI-TOF MS: A Reliable Tool in the Real Life of the Clinical Microbiology Laboratory" submitted for review in the journal Microorganisms describes the potential applications of MALDI MS technique in microbial identification. The authors have partially revised the manuscript; however, they have not yet addressed several issues raised in the previous review.

1. The term 'chemical matrix' should initially be elucidated as, for instance, 'the sample is mixed prior to measurement with a low-molecular-weight organic acid known as the matrix'.

2. The term 'Peptide Mass Fingerprint' is incorrect in the context of analyzing and identifying bacteria using MALDI. Although it has been used in this context in some articles, fundamentally, it is an analytical technique for protein identification, where an unknown protein of interest is first cleaved into smaller peptides, the absolute masses of which can be precisely measured using a mass spectrometer, such as MALDI-TOF.

3. The Shimadzu Axima is not a recently introduced system for microorganism identification. This system was introduced into use in the year 2011.

4. The Vitek MS and Shimadzu Axima systems are similar to each other, and the database is constructed in the same manner, which has been omitted in the description.

In conclusion, I kindly request the incorporation of appropriate revisions into the manuscript.

Author Response

Reviewer 2 Report (Previous Reviewer 2)

Comments and Suggestions for Authors

The authors addressed my comments. One more comment the review seems difficult to follow. I suggest the authors to include a table for summerizing the application of MALDI-TOF MS for microbes

Comments on the Quality of English Language

English is fine

Author Response

Reviewer 3 Report (New Reviewer)

Comments and Suggestions for Authors

The article under review is devoted to an actual topic the MALDI-TOF MS application in the diagnostic microbiology laboratory. Authors have carefully revised the wide range of scientific data and results presented in publications; the article provides a significant list of references.

 Overall, the article is relevant and advanced in the clinical laboratory diagnostics. However, for the successful publication of this manuscript, the following corrections must be made.

- As a general practice, when analyzing a great number of publications, the authors provide tables for convenient presentation. I recommend you to create a table in this manuscript, in which you would analyze the use of the MALDI-TOF MS l for studying different species of microorganisms with appropriate references.

- There are 2 figures in the manuscript, but I believe that these figures should be improved.

- One more illustration should be added to illustrate the basic steps of the sample pretreatments before the analysis.

Author Response

Reviewer 4 Report (New Reviewer)

Comments and Suggestions for Authors

This study summarized the MALDI-TOF MS in the clinical microbiology laboratory. This review is meaningful in this field. However, revisions needed to be performed before accepted for publication.

Main concerns:

(1)   What’s the advantage and disadvantage of MALDI-TOF MS compared to other clinical microbiology laboratory such as RT-PCR, the next generation sequence and biochemical detection.

(2)   The history applied in clinical microbiology laboratory was suggested to summarized in a figure to enhance article readability and citations

(3)   What’s the direction of future development in the application of MALDI-TOF MS in clinical microbiology laboratory

(4)   It was suggested to summarize the clinical microbiology which could be well detected by MALDI-TOF MS in a table.

Round 2

Reviewer 1 Report (Previous Reviewer 1)

Comments and Suggestions for Authors

The manuscript has been revised in accordance with the reviewers' suggestions and can be accepted for publication as is.

Reviewer 2 Report (Previous Reviewer 2)

Comments and Suggestions for Authors

No further comments

Comments on the Quality of English Language

Language is ok

This manuscript is a resubmission of an earlier submission. The following is a list of the peer review reports and author responses from that submission.

Round 1

Reviewer 1 Report

Comments and Suggestions for Authors

The manuscript titled "MALDI-TOF MS: A Reliable Tool in the Real Life of the Clinical Microbiology Laboratory" submitted for review in the journal Microorganisms describes the potential applications of MALDI MS technique in microbial identification. Unfortunately, the article exhibits numerous formal errors and does not contribute novelty to the presented field, as there are already several review articles covering similar topics. In my opinion, the presented manuscript is not suitable for publication, and I recommend its rejection in its current form.

The review paper concerns the application of the MALDI MS technique to identify bacteria in clinical samples.

The manuscript is not an original contribution to the field described, there are many review papers on similar topics, such as: doi: 10.3389/fmicb.2015.00791, 10.1016/j.jenvman.2021.114359, 10.3390/microbiolres14010008, 10.1186/s12879-019-4584-0, 10.1111/j.1574-6976.2011.00298.x, 10.1007/s00253-011-3783-4.

In the introduction, the authors should describe the MALDI MS technique, which is used for more than just microbial identification. The paper should include information on the ionization technique and sample preparation methods. 

- "chemical matrix" -> chemical compound called matrix - John Fenn received the Nobel Prize for the electrospray method, for the LDI technique for the analysis of macromolecular compounds actually received Koichi Tanaka, but he used cobalt nanoparticles as a matrix, it would be appropriate to mention here the pioneering work on the use of organic acids as matrices such as: doi: 10.1021/ac00171a028

- "MALDI requires a short pre-analytical step simply consisting in a mix prepared with an aliquot of the sample to be tested and an aliquot of a chemical matrix generating the ionizations of the molecules composing the sample; the ions generated by this chemical activity on the molecules present in the sample are analyzed measuring their mass-tocharge ratios (m/z) and from this the spectra are generated and interpreted by comparing with the spectra present in the data base of the spectrometer" - This sentence is incomprehensible. The ions are not generated by mixing the analyte with the matrix but by a laser pulse absorbed by the matrix. The compound used as a matrix should have the ability to absorb radiation of the wavelength emitted by the laser. In addition, the matrix's task is to protect the sample from direct laser impact which could lead to compound fragmentation.

- "Peptide Mass Fingerprint" (PMF) - This term is incorrect. Protein profiles, not peptide profiles, are used in the identification of microorganisms. - "...including specific spectra over 20,000 microbial species of medical interest..." - To the best of my knowledge, there is no spectrometer manufacturer with such a database of strains to date. To date, the largest database of bacterial protein profiles is Bruker Biotyper with about 11,000 strains.

- "...a score value of 2.0 is considered a reliable criterion of bacterial species identification albeit the spectra obtained from a single strain are not identical in repeated run because of a method-inherent noise. Sore values between 1.7 to 2.0 are considered reliable for genus identification" - This is true for the Bruker Biotyper or Zybio EXS2600 base, but the scoring is different for other spectrometers such as the VITEK MS or the aforementioned ASTA MicroIDSys. It would be necessary here to compare the databases of different manufacturers and the way the spectra are created in the databases (in the case of Bruker it is an average spectrum consisting of dozens of spectra of a given strain, while in the case of VITEK MS it is a so-called super spectrum averaging the results for many strains).

- The authors describe only three selected instruments for identifying microorganisms by MALDI, while there are many more available on the market: the Axima (Shimadzu, Japan) doi: 10.1016/j.mimet.2012.03.003, Autof MS100 (Autobio, China) doi: 10.1155/2021/6667623., or EXS2600 (Zybio, China) doi: 10.1128/jcm.01913-22, 10.1016/j.diagmicrobio.2023.116150. It would be appropriate to mention these identification systems.

Reviewer 2 Report

Comments and Suggestions for Authors

In the review,  the authors discussed the application of  MALDI-TOF MS. The authors discussed the prinicples of the procedure, application in identification of bacteria, fungi and antibiogram resistance.

In general the review is comperhensive, however it is not organoized 

Major comments

1) In section 4.1. Data base implementation; the author can organize data to Gm +ve bacteria, Gm-ve bacteria, Mycobacterium, other bacteria, fungi or mold, etc. Or they can include a table summerizing the main results and findings.

2) Limitations of MALDI-TOF MS  should be included as a separate title and paragraph

3) Future persepctives including the possibility of detection of viruses should be discussed

Comments on the Quality of English Language

Moderate language editing